# *Slc7a8* Deletion Is Protective against Diet-Induced Obesity and Attenuates Lipid Accumulation in Multiple Organs

**DOI:** 10.3390/biology11020311

**Published:** 2022-02-16

**Authors:** Reabetswe R. Pitere, Marlene B. van Heerden, Michael S. Pepper, Melvin A. Ambele

**Affiliations:** 1Institute for Cellular and Molecular Medicine, Department of Immunology and SAMRC Extramural Unit for Stem Cell Research and Therapy, Faculty of Health Sciences, University of Pretoria, Pretoria 0001, South Africa; reabetswe.pitere@tuks.co.za (R.R.P.); michael.pepper@up.ac.za (M.S.P.); 2Department of Oral Pathology and Oral Biology, School of Dentistry, Faculty of Health Sciences, University of Pretoria, Pretoria 0001, South Africa; marlene.vanheerden@up.ac.za

**Keywords:** obesity, adipose tissue, adipogenesis, lipid accumulation, SLC7A8, fat depots, tissue histology

## Abstract

**Simple Summary:**

The development of obesity can be attributed to adipocyte hypertrophy or hyperplasia which lead to increased adiposity. The C57BL/6 mouse is an excellent model to study metabolic syndromes often associated with obesity development. Mice fed on a high-fat diet are susceptible to weight gain, leading to the development of obesity and its associated metabolic syndrome. Here, we report findings from targeting a novel potential human adipogenic gene (*SLC7A8*) under conditions of obesity development using a mouse model of diet-induced obesity (DIO). The results indicate that deleting *slc7a8* in mice significantly protects against DIO and improves glucose metabolism. Deficiency in *slc7a8* was observed to significantly attenuate adipocyte hypertrophy in white and brown adipose tissue and to reduce lipid accumulation in many organs. Furthermore, inflammation was significantly reduced in the adipose tissue and liver of *slc7a8*-deficient mice with DIO. Overall, the results from this study show that *slc7a8* is an important molecular regulator of obesity development and mediates its function by reducing lipid accumulation in multiple organs. Hence, SLC7A8 could serve as a potential therapeutic target to combat the development of obesity and other pathophysiological conditions associated with excess lipid accumulation.

**Abstract:**

Adipogenesis, through adipocyte hyperplasia and/or hypertrophy, leads to increased adiposity, giving rise to obesity. A genome-wide transcriptome analysis of in vitro adipogenesis in human adipose-derived stromal/stem cells identified SLC7A8 (Solute Carrier Family 7 Member 8) as a potential novel mediator. The current study has investigated the role of SLC7A8 in adipose tissue biology using a mouse model of diet-induced obesity. *slc7a8* knockout (KO) and wildtype (WT) C57BL/6J mice were fed either a control diet (CD) or a high-fat diet (HFD) for 14 weeks. On the HFD, both WT and KO mice (WTHFD and KOHFD) gained significantly more weight than their CD counterparts. However, KOHFD gained significantly less weight than WTHFD. KOHFD had significantly reduced levels of glucose intolerance compared with those observed in WTHFD. KOHFD also had significantly reduced adipocyte mass and hypertrophy in inguinal, mesenteric, perigonadal, and brown adipose depots, with a corresponding decrease in macrophage infiltration. Additionally, KOHFD had decreased lipid accumulation in the liver, heart, gastrocnemius muscle, lung, and kidney. This study demonstrates that targeting *slc7a8* protects against diet-induced obesity by reducing lipid accumulation in multiple organs and suggests that if targeted, has the potential to mitigate the development of obesity-associated comorbidities.

## 1. Introduction

Obesity is characterised by an excess accumulation of adipose tissue when energy intake exceeds energy expenditure. The expansion of adipose tissue in obesity occurs either through adipocyte hyperplasia and/or hypertrophy. The result is dysfunctional adipose tissue mainly as a consequence of adipocyte hypertrophy, which leads to adverse metabolic consequences and chronic inflammation [1]. The distribution of adipose tissue in obesity plays an important role in the development of obesity-associated comorbidities. Accumulation of fat in intra-abdominal depots (visceral depots) gives rise to insulin resistance and is also associated with an increased risk of cardiovascular disease [2]. Subcutaneous white adipose tissue (WAT) is the most common adipose tissue in healthy lean individuals and serves as a metabolic sink for excess lipid storage [3]. Brown adipose tissue takes up fatty acids from the circulation to generate heat, which helps to clear plasma triglycerides, thereby reducing the accumulation of lipid in visceral depots [4]. In obesity, where the storage capacity of adipose tissue is exceeded because of an inability to either produce new adipocytes (limited hyperplasia) or expand further (limited hypertrophy), excess fat is redistributed to peripheral organs such as the liver and skeletal muscle, which increases the risk of metabolic-associated syndromes such as hyperglycaemia, hyperinsulinemia, atherosclerosis, dyslipidaemia, and systemic inflammation [3,5]. Hypertrophy in brown adipose tissue (BAT) may impair its function in acting as a sink for excess blood glucose and clearing free fatty acids from circulation, thereby contributing to the development of insulin resistance and hyperlipidaemia in obesity [3]. Therefore, mitigating adipocyte hypertrophy in both WAT and BAT depots is paramount to improving metabolic health.

Inflammation is a key consequence of adipose tissue expansion that occurs during weight gain and contributes to the development of chronic low-grade systemic inflammation as seen in obesity. This expansion of adipose tissue is characterised by increased infiltration of immune cells, with a predominance (around 60%) of macrophages, in response to chemokines that are produced by hypertrophic adipocytes [6]. The majority are derived from circulating monocytes, with a small proportion coming from the proliferation of adipose tissue-resident macrophages [7]. Tissue-resident macrophages present in normal or lean adipose tissue are of the M2 anti-inflammatory macrophage phenotype, which expresses markers such as the mannose receptor (CD206) and is thought to be responsible for maintaining tissue homeostasis [8]. Macrophage infiltration in adipose tissue appears as crown-like clusters, which are believed to signify an immune response to dying or dead adipocytes [9]. These infiltrating macrophages undergo a phenotypic switch to an M1 proinflammatory phenotype [10].

Several studies have suggested countering the process of fat cell formation (adipogenesis) to combat obesity development. This has led to several molecular determinants being described as playing an important role in adipogenesis [11]. Except for PPARɣ [12,13], molecular determinants of adipogenesis have proven to be of limited clinical utility. Therefore, more research is needed to identify new molecular determinants of adipogenesis, which could play a role in obesity development and serve as potential therapeutic targets. We previously undertook a comprehensive, unbiased transcriptomic analysis of human adipose-derived stromal/stem cells undergoing adipogenesis in vitro, and identified several novel genes and transcription factors with possible roles in this process [14,15]. One of the novel genes identified was *SLC7A8* (Solute Carrier Family 7 Member 8), not previously described in the context of adipogenesis and/or obesity, which was significantly upregulated in the early phase of adipogenesis and declined significantly as the process progressed [14]. This suggested a role for this gene in the early stages of adipogenesis as a potential driver of adiposity and consequently obesity.

The *SLC7A8* gene encodes large neutral amino acid transporter small subunit 2 (LAT2) comprised of 535 amino acids, and it is located on chromosome 14q11.2 [16,17]. The gene is highly expressed in the kidneys and intestines [17]. Mutations in *SLC7A8* have been implicated in age-related hearing loss [18] and aminoaciduria [19]. Furthermore, SLC7A8 has been reported to be highly expressed in oestrogen receptor-positive breast cancer [20] and has also been implicated in cataract formation when defective [21]. Since SLC7A8 has not been described in the context of obesity or adipogenesis, the aim of this study was to investigate the functional role of *SLC7A8* in weight gain/obesity and lipid accumulation in various tissues and organs (such as perigonadal, mesenteric, inguinal subcutaneous, and interscapular brown adipose tissues and the liver, kidneys, heart, brain, lungs, and gastrocnemius muscle) using a mouse model of diet-induced obesity. Macrophage infiltration profiling in some of these tissues (i.e., perigonadal, mesenteric, inguinal subcutaneous, and interscapular brown adipose tissues and the liver, kidneys, and gastrocnemius muscle) was also performed.

## 2. Materials and Methods

### 2.1. Animals

This study was approved by the Research Ethics Committee, Faculty of Health Sciences and the Animal Ethics Committee, University of Pretoria (Ref. No.: 474/2019).

*Slc7a8* heterozygous (B6.129P2-Slc7a8tm1Dgen/J, #005842) C57BL/6J and inbred wildtype C57BL/6J mating pairs obtained from The Jackson Laboratory (Jackson Laboratory, Bar Harbour, ME, USA) were used to generate *Slc7a8* wildtype (WT) and knockout (KO) genotypes. Genotypes were confirmed by PCR (Appendix A). Both WT and KO mice were fed either a high-fat diet (HFD; D12492) or a control diet (CD; D12450J) from Research Diets, Inc. (Research Diets, Inc., New Brunswick, NJ, USA) for a period of 14 weeks, with termination timepoints at weeks 5 and 14. Weekly measurements were made of weight, food consumption and calorie intake. Unless otherwise stated, the nomenclature used for the different genotypes on either a CD or HFD for 14 weeks is WTCD (wildtype mice on control CD), WTHFD (wildtype mice on HFD), KOCD (*Slc7a8* Knockout mice on control CD), and KOHFD (*Slc7a8* knockout mice on HFD).

### 2.2. Glucose Tolerance and Insulin Sensitivity Tests

Glucose tolerance tests (GTT) and insulin sensitivity tests (IST) were performed in both KO and WT mice prior to introducing them to either CD or HFD. Mice were fasted for 4 h, and the baseline glucose concentration was measured. A 45% D-(+)- glucose solution (G8769) (Sigma-Aldrich, St. Louis, MO, USA) at 1.5 mg/g body weight and an insulin solution (I9278) (Sigma-Aldrich, St. Louis, MO, USA) at 0.8 mU/g body weight were then administered interperitoneally for GTT and IST, respectively. Blood from the tail vein was used to measure glucose concentration at 15, 30, 60, 90, and 120 min using an Accu-Check Instant Blood Glucose Meter (Roche Diagnostics, Basel, Switzerland).

### 2.3. Histology and Immunohistochemistry of Mouse Tissues and Organs

Mice on either CD or HFD were euthanised at weeks 5 and 14. This was followed by the collection of white adipose tissue from the inguinal (iWAT), perigonadal (pWAT), and mesenteric (mWAT) depots; interscapular brown adipose tissue (BAT); and the liver, kidneys, heart, brain, lungs, and gastrocnemius muscle. Ten percent formalin fixed paraffin embedded (FFPE) tissue sections were processed for histological analysis.

FFPE tissue sections were cut using a microtome and baked at 62 °C for 20 min followed by haematoxylin and eosin (H&E) staining using a Leica Autostainer XL (Leica Microsystems, Wetzlar, Germany). Slides were mounted with DPX (distyrene, plasticiser, xylene) and imaged using an Axiocam 305 colour microscope camera (Carl Zeiss Meditec AG, Jena, Germany) and ZEN 2.6 blue edition software (Carl Zeiss Meditec AG, Jena, Germany). In the absence of quantification, representative images are shown.

Immunohistochemical analysis of macrophages was performed as previously described [22]. FFPE sections were baked overnight at 54 °C, followed by dewaxing in xylene. The sections were then hydrated through a series of ethanol concentrations, rinsed with distilled water, and treated with 3% hydrogen peroxide for 5 min at 37 °C. Heat-induced epitope retrieval was performed in citrate buffer pH 6,1 using the Dako Target Retrieval Solution S1699 (Dako, Carpinteria, CA, USA) and a 2100 Retriever Unit (Electron Microscopy Sciences, Washington, PA, USA). The sections were rinsed in PBS/Tween buffer and treated with 5% Normal Goat Serum (Dako X0907) for 30 min, after which they were incubated overnight at 4 °C with a 1:25 dilution of F4/80 monoclonal rat anti mouse antibody BM8 (#14-4801-82) (Thermo Fisher Scientific, Bedford, MA, USA) or 1:100 F4/80 rat anti mouse antibody clone A3-1 (Bio-Rad Laboratories, Irvine, CA, USA). The sections were rinsed in PBS/Tween buffer before being incubated for 60 min in 1:200 goat anti rat IgG (H+L) antibody conjugated to horseradish peroxide (HRP) (#31470) (Invitrogen, Bedford, MA, USA). The slides were then developed in 3,3′ diaminobenzidine (DAB) chromogen to visualise F4/80 protein staining. All images were taken and analysed at 20× magnification.

### 2.4. Statistical and Image Analyses

Images from H&E and immunohistochemical staining were analysed using ImageJ Fiji (https://imagej.nih.gov/ij/download.html, accessed on 5 May 2021) or Aperio ImageScope version 12.4.3.5008 software (Leica Biosystems, Wetzlar, Germany). Morphometric analysis of the various tissue sections was estimated by measuring the diameter of at least 120 cells distributed across the tissue viewed on a single microscope slide. Semiquantitative analysis of F4/80 staining using ImageJ Fiji was conducted according to the protocol described by Crowe and Jue, [23] to quantify macrophages in the tissues.

Statistical analyses were conducted using GraphPad Prism 5 (GraphPad Software Inc., San Diego, CA, USA). Values are expressed as mean ± SEM. One-way ANOVA followed by Bonferroni corrections was used to compare means among three or more categories. When comparing two means, a two-tailed unpaired Student’s t-test was used. Two-way ANOVA with Bonferroni corrections was used where necessary. Statistically significant results are indicated as * *p* < 0.05, ** *p* < 0.01, *** *p* < 0.001.

## 3. Results

### 3.1. Deficiency in Slc7a8 Protects against Diet-Induced Obesity

WT and KO mice gained significantly more weight at 14 weeks on HFD than did WTCD (*p* < 0.05 to *p* < 0.001) and KOCD (*p* < 0.05 to *p* < 0.001), respectively (Figure 1a). No significant differences were observed between WT and KO mice on CD. Interestingly, KOHFD gained significantly (*p* < 0.05 to *p* < 0.001) less weight than WTHFD, which was evident from week 3 (Figure 1a). Significant weight gain in WTHFD was associated with significantly larger (*p* < 0.001) iWAT, pWAT, mWAT, BAT, and liver compared with WTCD and KOHFD. Only the pWAT of KOHFD was significantly larger than in KOCD14 (Figure 1d). WTHFD and KOHFD mice appeared visibly larger in size when compared to their respective lean counterparts (Appendix A). Total food consumption of WTHFD was significantly greater than that of WTCD (*p* < 0.001) and KOHFD (*p* < 0.01) (Figure 1b). Energy intake increased significantly between WTCD and WTHFD (*p* < 0.01 at week 5 and *p* < 0.001 from week 6 to week 14) and between KOCD and KOHFD (*p* < 0.05 at week 3, *p* < 0.01 at week 4, and *p* < 0.001 from week 5 to 14) (Figure 1c). A significant difference (*p* < 0.001) in cumulative caloric intake was observed between WTHFD and KOHFD from week 9 to week 14. No significant differences in calorie intake were seen between KOCD and WTCD.

### 3.2. Deficiency in Slc7a8 Has No Effect on Glucose and Insulin Metabolism but Significantly Improves Glucose Tolerance When on HFD

WT and KO mice showed no difference in metabolism of exogenous glucose (Figure 2a,b) or insulin (Figure 2c,d) prior to introducing them to the experimental diets. However, significantly elevated glucose levels (*p <* 0.01) were observed for the KO mice at 30 min.

After 5 weeks on experimental diets, no significant difference was observed in glucose metabolism between WT and KO on either the CD or HFD (Figure 3a,b). However, at 14 weeks, WTHFD had significantly higher glucose levels than KOHFD and WTCD starting from 30 min (Figure 3c). Although WTHFD had a larger AUC than KOHFD and WTCD, this was not statistically significant (Figure 3d). No significant differences were observed between the AUCs of WTCD5, WTHFD5, KOCD5, and KOHFD5 when compared to their 14-week counterparts (WTCD14, WTHFD14, KOCD14, and KOHFD14; Figure 3b,d).

### 3.3. Slc7a8 Deletion Attenuates Adipocyte Hypertrophy in White and Brown Adipose Depots

Two representative microscopic images of each adipose tissue depot are depicted in Figure 4. The pWAT from WTCD (Figure 4a) and KOHFD (Figure 4b) had significantly smaller (*p* < 0.001) adipocytes (A) than did WTHFD (Figure 4c), as indicated in Figure 4d. The number of adipocytes per field was significantly higher in WTCD (*p* < 0.001) and KOHFD (*p* < 0.05) than in WTHFD (Figure 4e). The iWAT in WTHFD (Figure 4h) had a significant increase (*p* < 0.001) (Figure 4i) in adipocyte hypertrophy compared with that in KOHFD (Figure 4g) and WTCD (Figure 4f). Similarly, a significant increase (*p* < 0.001) was observed in the adipocyte size of mWAT from WTHFD (Figure 4m) in comparison with those from KOHFD (Figure 4l) and WTCD (Figure 4k), Figure 4n. The number of adipocytes per field was significantly lower in the mWAT (*p* < 0.01) (Figure 4j) and iWAT (*p* < 0.001) (Figure 4o) of WTHFD than in those of WTCD, as well as in the mWAT (*p* < 0.01) and iWAT (*p* < 0.001) of WTHFD than in those of KOHFD. Lipid droplet accumulation was greater in the BAT of WTHFD (Figure 4r) than in that of WTCD (Figure 4p) and KOHFD (Figure 4q). Additionally, as early as 5 weeks on the experimental diet, adipocyte hypertrophy was greater in WTHFD than in KOHFD and WTCD in pWAT, mWAT, and iWAT, and larger lipid droplets were observed in the BAT of WTHFD (Appendix A).

### 3.4. Deletion of Slc7a8 Reduces Liver Steatosis in Diet-Induced Obese Mice

Liver sections from WTHFD (Figure 5c) showed lipid accumulation, which could be categorised as microvesicular (circled, Figure 5c) and macrovesicular (indicated in black arrow, Figure 5c) steatosis. This phenomenon was absent in liver sections of WTCD (Figure 5a), while macrovesicular steatosis observed in KOHFD (Figure 5b) had smaller lipid droplets than that in WTHFD (Appendix A).

### 3.5. Deficiency in Slc7a8 Decreases Lipid Accumulation in Gastrocnemius Muscle

Skeletal muscle myocyte atrophy was observed in WTHFD (Figure 6c), which had significantly smaller myocytes (*p* < 0.001) (Figure 6g) than WTCD (Figure 6a). The deletion of *slc7a8* increased myocyte size in KOHFD (Figure 6b) compared to WTHFD (Figure 6g). Accumulation of perimuscular adipose tissue (PMAT) (Figure 6f) was greater in WTHFD than in KOHFD (Figure 6e) and WTCD (Figure 6d). At week 5, the KOHFD had significantly larger myocytes (*p* < 0.001) and less adipose tissue than WTHFD (Appendix A).

### 3.6. Deficiency in Slc7a8 Reduces Accumulation of Epicardial Adipose Tissue

The increase in the accumulation of epicardial adipose tissue (EAT—white adipose tissue) observed in WTHFD (Figure 7c) compared to WTCD (Figure 7a) was decreased following the deletion of *slc7a8*, KOHFD (Figure 7b). Larger lipid droplets were observed in the brown/beige adipose tissue (a property of epicardial adipose tissue) in WTHFD (Figure 7f) than in that in WTCD (Figure 7d) and KOHFD (Figure 7e).

### 3.7. Deficiency in Slc7a8 Reduces Lipid Accumulation in the Ganglion Layer in Diet-Induced Obese Mice

In the cerebral cortex, lipid droplets were seen in KOHFD (Figure 8b) and WTHFD (Figure 8a), but not in WTCD (Figure 8c).

### 3.8. Deficiency in Slc7a8 Reduces Lipid Accumulation in the Kidney

Lipid droplet accumulation was reduced in KOHFD (Figure 9b) to the level observed in WTCD (Figure 9a). WTHFD showed visibly larger lipid droplets (Figure 9c).

### 3.9. Deficiency in Slc7a8 Reduces Adipose Tissue Accumulation in the Lungs

Histological analysis of the lungs showed greater accumulation of adipose tissue in WTHFD (Figure 10c) than in WTCD (Figure 10a). The accumulation of adipose tissue in DIO appeared to be reduced in KOHFD (Figure 10b). Lipid accumulation in the lungs was observed as early as week 5, with more adipose tissue in WTHFD and KOHFD than in WTCD (Appendix A).

### 3.10. Deficiency in Slc7a8 Reduces White Adipose Tissue Inflammation in DIO

Immunohistochemical staining for F4/80, a mouse macrophage marker, was performed to assess the presence of macrophages in pWAT, mWAT, iWAT and brown adipose tissue. Deletion of *slc7a8* significantly decreased macrophage infiltration (indicated by black arrows) in the pWAT (Figure 11c; *p* < 0.01), mWAT (Figure 11h; *p* < 0.05), and iWAT (Figure 11k; *p* < 0.01) of KOHFD (Figure 11a,f,i) compared with those of WTHFD (Figure 11b,g,j). No significant difference was observed in the brown adipose macrophage inflammation profile (Figure 11n) between WTHFD (Figure 11m) and KOHFD (Figure 11l). Figure 11d,e shows the negative controls for KOHFD and WTHFD, respectively, in perigonadal adipose tissue.

### 3.11. Deficiency in Slc7a8 Reduces Inflammation in the Liver

Deficiency in *slc7a8* resulted in a significant (*p* < 0.05) (Figure 12c) reduction in macrophages in the liver from KOHFD, Figure 12a, compared to WTHFD (Figure 12b). Figure 12d,e show negative control staining of KOHFD and WTHFD, respectively; no staining for macrophages was detected in these controls.

### 3.12. Deficiency in Slc7a8 Has No Effect on the Presence of Macrophages in the Kidney or Gastrocnemius Muscle in DIO

The presence of macrophages in the kidney of KOHFD (Figure 13a) and WTHFD (Figure 13b) was similar (Figure 13c). This observation was the same for the gastrocnemius muscle of KOHFD (Figure 13d) and WTHFD (Figure 13e), with no statistical difference in macrophage profile between them (Figure 13f).

## 4. Discussion

Obesity is characterised by excessive accumulation of adipose tissue, and is associated with the development of metabolic syndromes affecting many organs and tissues in the body. The search for molecular factors that play a role in attenuating lipid accumulation in conditions such as diet-induced obesity is paramount to identifying good candidates for therapeutic interventions that mitigate the development of obesity associated comorbidities. Studies of adipogenesis in human-derived stromal/stem cells in vitro have served as an excellent model for identifying molecular factors with a potential role in adipocyte formation and lipid accumulation/metabolism [11,14]. This study investigated the role of a previously identified novel human adipogenic gene, SLC7A8 [14], in diet-induced obesity and its effect on adipose tissue accumulation in different organs and tissues. To achieve this, *slc7a8* knockout (KO) and wildtype (WT) C57BL/6 mice were fed either a HFD or a nutrient-matched CD for 14 weeks; this was followed by analyses of different parameters.

Weight gain, food, and caloric intake between WTCD and KOCD were similar, indicating that *slc7a8* deletion had no effect on food intake, caloric consumption, or weight gain on a normal diet. WTHFD gained significantly more weight (*p* < 0.001) than WTCD starting from week 3 (Figure 1a), with a significantly higher caloric intake (*p* < 0.01 to *p* < 0.001) than WTCD (Figure 1c). Total food consumption was not significantly different during the 14-week period except at week 8, where food consumption in WTHFD was significantly higher (*p* < 0.05). This indicates that the occurrence of diet-induced obesity was due to an increase in caloric intake when on HFD. Interestingly, the *slc7a8*-deficient genotype on HFD (KOHFD) gained significantly less weight (*p* < 0.05 to *p* < 0.001) than the WTHFD starting from week 3 (Figure 1a). This suggests that *slc7a8* deletion was protective against diet-induced obesity. The significant decrease in weight gain in KOHFD was accompanied by significantly lower tissue mass of iWAT, mWAT, pWAT, BAT, and liver compared with those in WTHFD (Figure 1d). Strikingly, it was observed that KOHFD gained significantly more weight (*p* < 0.05 to *p* < 0.001) than KOCD from week 8, and this corresponded to a significantly larger pWAT in KOHFD than in KOCD (Figure 1d). This indicates that weight gain by KOHFD was due to pWAT expansion and suggests that pWAT was the primary site of lipid accumulation in the KO phenotype.

BAT in WTHFD (Figure 4r) displayed enlarged lipid droplets compared with those in WTCD (Figure 4p). A recent study showed that following 20 weeks of feeding mice on an HFD, lipid accumulation did not influence the function of brown adipose tissue. However, the authors speculated that if the period of HFD feeding were to be extended, a malfunction of BAT would be observed in obese mice [24]. We observed in the current study that KOHFD (Figure 4q) attenuated adipocyte hypertrophy and lipid accumulation in BAT. This suggests that *slc7a8* deletion could be protective against the long-term effects of BAT hypertrophy and malfunctioning caused by DIO.

Furthermore, it was observed that WTHFD had a significantly greater caloric intake than KOHFD (Figure 1c) while food consumption was similar, except at week 11, when a significant difference (*p* < 0.05) was observed. It is possible that the deletion of *slc7a8* regulated weight gain on HFD by burning calories quicker than WTHFD, since both KOHFD and WTHFD had similar caloric intake up to week 8 (Figure 1c), but as early as week 5, adipocyte hypertrophy was already significantly greater in WTHFD than in KOHFD (Appendix A). Additionally, food and caloric intake was similar between KO and WT on a normal diet, with differences observed only on HFD; this could suggest satiety in KOHFD, as caloric intake significantly decreased after week 8 (Figure 1c).

Adipose tissue expansion in obesity is commonly associated with conditions such as hyperglycaemia, impaired glucose tolerance, and insulin resistance [25]. To investigate the effect of *slc7a8* deletion on the metabolism of exogenous glucose and insulin, GTT and IST were performed on all animals (KO and WT) prior to introducing them to an experimental diet (Figure 2a,b). Importantly, there was no significant difference between the KO and WT mice for either test. This shows that the deletion of *slc7a8* had no effect on their ability to metabolise glucose and insulin efficiently. However, significantly higher levels of blood glucose were seen in KO mice at 30 min of the GTT (Figure 2a), which later returned to normal, without any change in the AUC between KO and WT (Figure 2b). Both WTCD and KOCD at 5 and 14 weeks showed similar trends in glucose metabolism (Figure 3a,c), with no difference in the AUC (Figure 2b,d), suggesting that glucose metabolism was unaltered in *slc7a8*-deficient mice on a normal diet. Under conditions of DIO, WTHFD showed significantly higher levels of glucose intolerance than WTCD, and this effect was significantly improved in KOHFD, with blood glucose levels returning to baseline levels at the end of the GTT (Figure 3c). This demonstrated that *slc7a8* deletion significantly improved glucose metabolism in DIO.

WTHFD showed significantly larger adipocytes in the pWAT, mWAT, and iWAT (Figure 4) than WTCD. The adipose tissue hypertrophy in WTHFD may increase susceptibility to hyperglycaemia. In an obese phenotype, insulin signalling is usually impaired, which results in reduced glucose uptake by muscles and thus increased glucose levels in the circulation [26]. pWAT was significantly larger (*p* < 0.001) than iWAT and mWAT in WTHFD (Appendix A), which may be suggestive of pWAT being the main site of lipid accumulation in this group, as was observed in the KO group. Abdominal/visceral obesity is critical to the development of metabolic syndrome, and accumulation of adipose tissue in the abdomen correlates with metabolic syndrome more than lipid accumulation in the subcutaneous depot [27]. The larger pWAT in WTHFD may thus have been responsible for the glucose intolerance observed in these mice. Lipid accumulation in the liver presented as microvesicular steatosis (characterised by small lipid droplets in the cytoplasm of hepatocytes) and macrovesicular steatosis (large lipid droplets) (Figure 5), which are both important in the development of nonalcoholic fatty liver disease (NAFLD) [28,29], and was observed in WTHFD but not in WTCD. The presence of lipid droplets in WTHFD liver could be due to the redistribution of excess lipids to peripheral organs, such as the liver or muscles as seen in the obese phenotype, when the storage capacity of adipose tissue is exceeded [3,5]. The liver has previously been reported to be the major site for storage of free fatty acids (FFA) released from white adipose tissue in an obese phenotype [30]. Furthermore, the vast majority of hepatic triglycerides in obese individuals with NAFLD are from FFA released from adipose tissue [31]. The observations made in our study indicated that KOHFD attenuated both the macrovesicular and the microvascular steatosis seen in WTHFD, suggesting that *slc7a8* deletion could be protective against NAFLD in DIO.

DIO is often associated with the recruitment and accumulation of macrophages in adipose depots. The F4/80 antibody is a marker for macrophages in mouse tissues [10,32] and was utilised in this study. Adipose tissues from obese WTHFD showed significantly more macrophages, which indicated increased inflammation when compared with KOHFD (Figure 11). Thus, *slc7a8* deletion significantly improved the inflammatory profile of adipose tissues in DIO. The liver tissue sections in WTHFD showed significantly elevated levels of macrophages. The observed histopathological changes in the liver that occurred because of DIO were improved by *slc7a8* deletion in KOHFD (Figure 12).

Apart from metabolic syndromes that are associated with excess adipose tissue accumulation, obese individuals are also prone to developing pulmonary disorders such as chronic obstructive pulmonary disease (COPD) or asthma [33]. In DIO, the lungs of WTHFD showed an increase in adipose tissue accumulation, which was reduced in KOHFD (Figure 10). A previous study showed that accumulation of adipose tissue in the lungs increased with an individual’s body mass index (BMI) [34]. Additionally, an increase in adipose tissue affects the structure of the lungs, resulting in the blockage of airways and causing inflammation, which ultimately gives rise to pulmonary disease [33,34]. We observed that the deletion of *slc7a8* attenuated adipose tissue accumulation in DIO; this could mitigate the development of obesity-associated lung pathologies.

DIO resulted in a significant reduction in gastrocnemius muscle myocyte size in WTHFD compared with that in WTCD, and the deletion of *slc7a8* decreased this effect of DIO (KOHFD) on myocyte size (Figure 6g). Additionally, perimuscular adipose tissue accumulation, which was observed to increase in the muscles of WTHFD, decreased in KOHFD (Figure 6e,f). Perimuscular adipose tissue has previously been shown to promote age- and obesity-related muscle atrophy by increasing muscle senescence [35]. Hence, the decrease in lipid accumulation due to *slc7a8* deletion in our study suggests an improvement in DIO-associated muscular disease. Conversely, there was no significant difference in the gastrocnemius muscle macrophage profile between WTHFD and KOHFD.

The development of cardiovascular diseases is associated with an increase in adiposity [36]. In DIO, the hearts of WTHFD showed greater accumulation of epicardial adipose tissue, which was found to decrease in the absence of *slc7a8* (Figure 7). Epicardial adipose tissue is located between the myocardium and epicardium and has properties of brown or beige adipose tissue. It is important for maintaining energy homeostasis and thermoregulation of the heart [37]. However, accumulation of epicardial adipose tissue is associated with increasing BMI and poses a risk for the development of cardiovascular disease [36].

Renal injury and disease have been associated with obesity, and studies in mice have documented renal morphological changes due to HFD [22,38]. An accumulation of lipid droplets in the kidneys was observed in DIO in WTHFD, suggesting that an increase in body weight could contribute to renal abnormalities. Lipid accumulation was reduced in KOHFD (Figure 9), suggesting that *slc7a8* deletion may improve kidney health in DIO.

## 5. Conclusions

This study has demonstrated that deletion of *slc7a8* in mice is protective against DIO by significantly reducing adipose tissue mass and lipid accumulation in multiple organs and tissues, and results in improved glucose tolerance in diet-induced obesity. Furthermore, our histological findings revealed that the negative effects of DIO on different organs and tissues are improved with *slc7a8* deletion, suggesting that this gene may contribute to the development of some obesity-associated comorbidities. Overall, the results from this study suggest that *slc7a8* might be a potential therapeutic target for controlling DIO, as well as for mitigating the development of some of the pathophysiological conditions associated with obesity. Nevertheless, further studies will be required to provide additional knowledge on how *slc7a8* regulates plasma parameters such hormones, lipids, and the cytokine inflammatory profile in DIO to reduce lipid accumulation in multiple organs and tissues.

## Figures and Tables

**Figure 1 biology-11-00311-f001:**
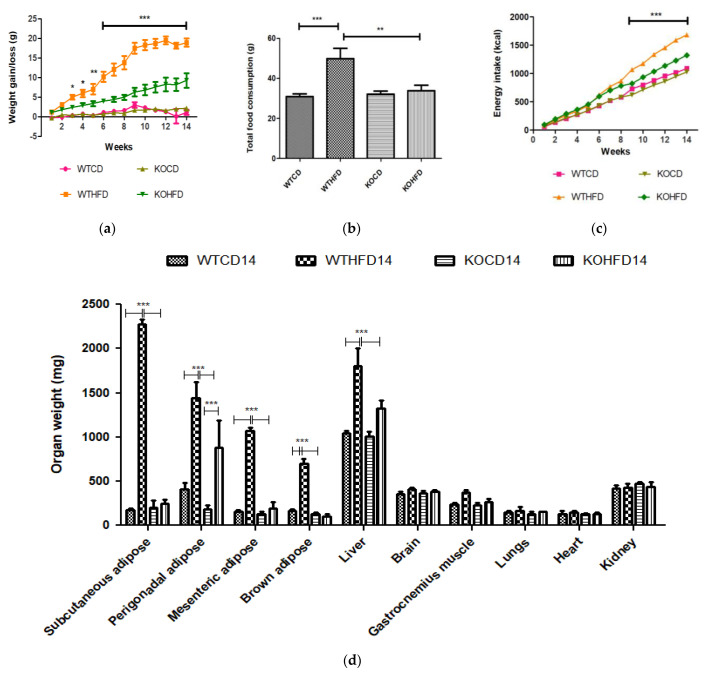
Effect of *slc7a8* deletion on body weight and caloric intake. (**a**) WTHFD gained significantly more weight throughout the 14-week period starting from week 2 than did WTCD (*p* < 0.05 to *p* < 0.001). KOHFD gained significantly more weight than KOCD (*p* < 0.05). The difference in weight gain between WTHFD and KOHFD was significant starting in week 3, with the *p*-value decreasing gradually from *p* < 0.05 to *p* < 0.001. WTCD and KOCD showed no differences in weight. (**b**) Total cumulative food consumption was significantly greater in WTHFD than in WTCD (*p* < 0.001) and KOHFD (*p* < 0.01). Energy intake increased significantly (from *p* < 0.01 at week 5 to *p* < 0.001 from week 6 to week 14) between WTCD and WTHFD. (**c**) A significant difference (*p* < 0.001) in caloric intake was observed between WTHFD and KOHFD from week 11 to week 14. Comparisons between KOCD and KOHFD revealed significant differences in calorie intake, with *p* < 0.05 at week 3, *p* < 0.01 at week 4, and *p* < 0.001 from week 5 to 14. No significant differences in caloric intake were seen between KOCD and WTCD. (**d**) WTHFD showed significantly larger (*p* < 0.001) iWAT, pWAT, mWAT, BAT, and liver compared to WTCD and KOHFD. Week 1–5: N = 18 for WTCD, WTHFD, and KOHFD and N = 17 for KOCD; week 6–8: N = 12 for WTCD, WTHFD, and KOHFD and N = 11 for KOCD; week 9–12: N = 6 for WTCD, WTHFD, and KOHFD and N = 5 for KOCD; week 13–14: N = 5 for WTCD, WTHFD, and KOCD and N = 6 for KOHFD. * *p* < 0.05, ** *p* < 0.01, *** *p* < 0.001.

**Figure 2 biology-11-00311-f002:**
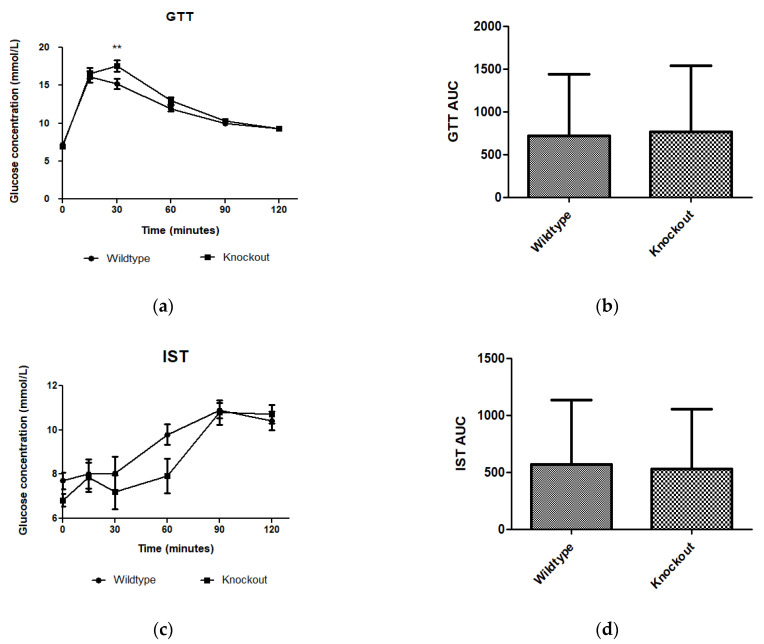
Effect of genotype on glucose tolerance and insulin sensitivity tests. GTT and IST were conducted before introducing the C57BL/6J wildtype and *Slc7a8* knockout mice to CD or HFD. (**a**,**c**) No significant differences were observed in GTT or IST between the WT and KO mice except for significantly higher glucose levels (*p* < 0.01) observed for the KO mice at 30 min with the GTT. (**b**,**d**) No significant differences were observed between the areas under the curve (AUCs) for WT and KO mice during GTT and IST. GTT: N = 47 for WT and N = 48 for KO; IST: N = 47 for WT and N = 44 for KO. ** *p* < 0.01.

**Figure 3 biology-11-00311-f003:**
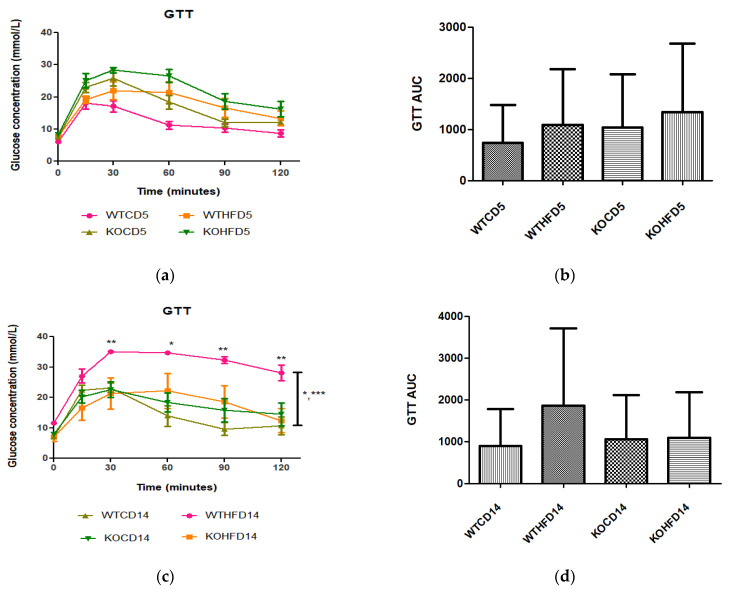
Glucose tolerance and insulin sensitivity tests of animals on experimental diet. (**a**) No significant differences were observed after experimental feeding between WT and KO mice on either CD or HFD at 5 weeks. (**c**) WTHFD showed significantly higher glucose levels than KOHFD (*p* < 0.05, 0.01) and WTCD (*p* < 0.05, 0.001) at 14 weeks. (**b**,**d**) No significant differences were observed between WTCD5, WTHFD5, KOCD5, and KOHFD5 and their respective 14-week counterparts. N = 6 for WTCD5, WTHFD5, KOCD5, KOHFD5, WTCD14 WTHFD14, and KOCD14 and N = 5 for KOHFD14. * *p* < 0.05, ** *p* < 0.01, *** *p* < 0.001.

**Figure 4 biology-11-00311-f004:**
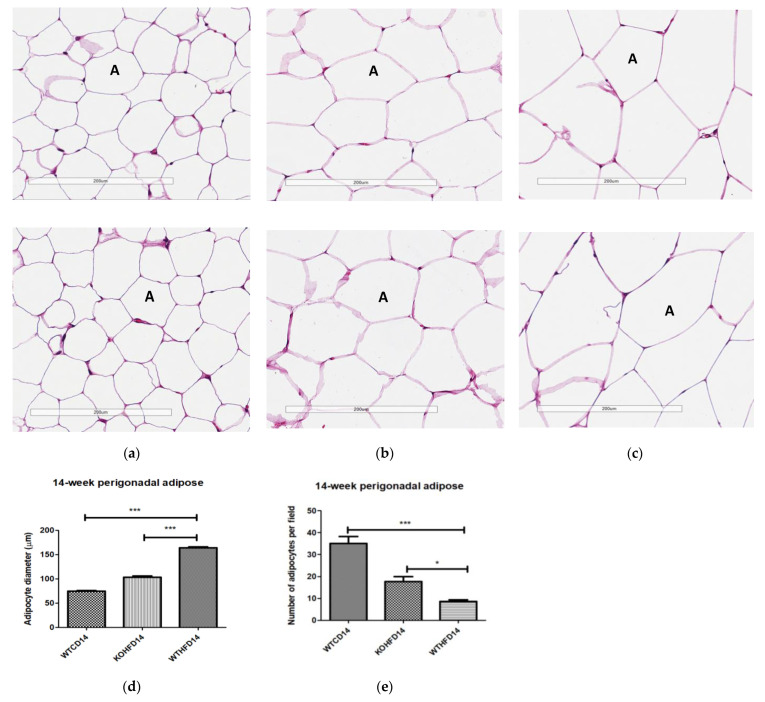
Adipocyte size distribution across the various adipose tissue depots. H&E-stained sections of perigonadal WAT (pWAT) (**d**) revealed that the WTHFD (**c**) had significantly larger (*p* < 0.001) adipocytes than WTCD (**a**) and KOHFD (**b**). The number of adipocytes per field was significantly lower in WTHFD than KOHFD (*p* < 0.05) and WTCD (*p* < 0.001), (**e**). Similarly, the adipocyte diameter in the inguinal subcutaneous WAT (iWAT) of WTHFD (**h**) was significantly greater (*p* < 0.001), (**i**), than that of WTCD (**f**) and KOHFD (**g**). Conversely, the number of adipocytes per view was significantly lower (*p* < 0.001) in WTHFD than in WTCD and KOHFD, (**j**). Significantly more (*p* < 0.001) adipocyte hypertrophy (**n**) was also observed in the mWAT of WTHFD, (**m**), than in WTCD, (**k**) and KOHFD (**l**). Additionally, significantly (*p* < 0.01) fewer adipocytes were viewed per field in WTHFD than in WTCD and KOHFD (**o**). Sections of the BAT revealed that WTCD (**p**) and KOHFD (**q**) had smaller lipid droplets compared with those observed in WTHFD (**r**). Magnification = 20× Scale bar = 200 µm. Key: A = adipocyte. N = 120 adipocytes. * *p* < 0.05, ** *p* < 0.01, *** *p* < 0.001.

**Figure 5 biology-11-00311-f005:**
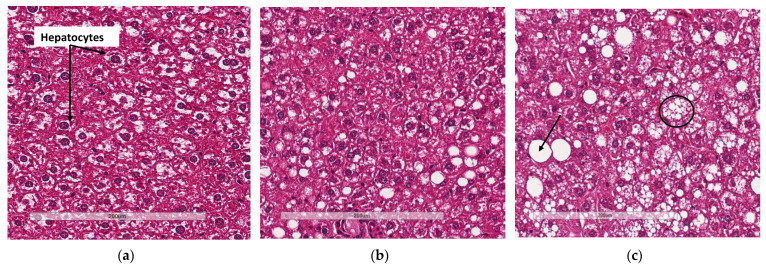
Lipid accumulation in the liver. H&E-stained liver sections showed the presence of micro- and macrovesicular steatosis in WTHFD (**c**), which was reduced in KOHFD (**b**) and absent in WTCD (**a**). Magnification = 20×. Scale bar = 200 µm.

**Figure 6 biology-11-00311-f006:**
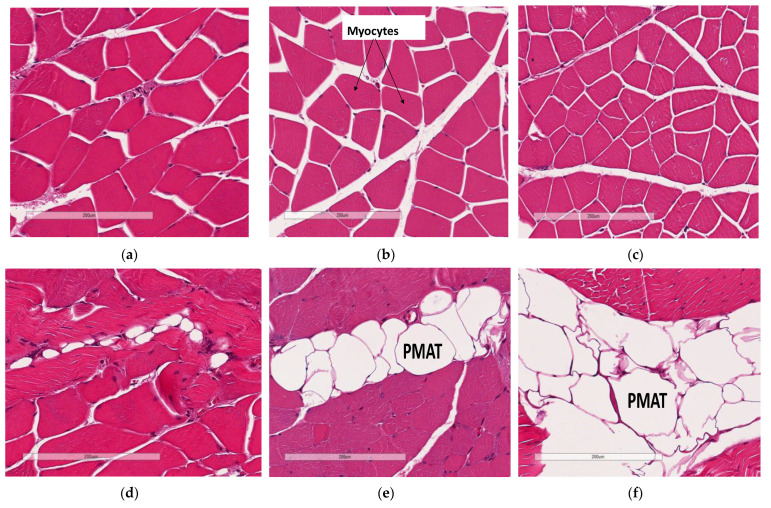
Effect of *slc7a8* deletion on adipose tissue accumulation and myocyte atrophy in gastrocnemius muscle. Myocyte atrophy was observed in WTHFD (**c**) when compared to WTCD (**a**) and KOHFD (**b**). WTHFD had significantly smaller (*p* < 0.001) (**g**) myocytes than WTCD. Greater perimuscular adipose tissue (PMAT) accumulation was seen in WTHFD (**f**) than in WTCD (**d**) and KOHFD (**e**). N = 120 myocytes. *** *p* < 0.001.

**Figure 7 biology-11-00311-f007:**
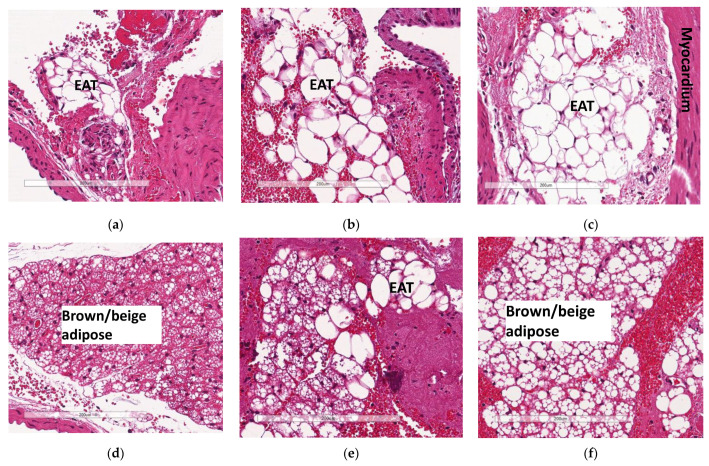
Effect of *slc7a8* on epicardial adipose tissue accumulation in the heart. H&E-stained heart sections showed a greater accumulation of epicardial adipose tissue, seen as a brown/beige adipose depot, in the WTHFD (**c**) compared to WTCD (**a**) and KOHFD (**b**). The images demonstrate that the WTHFD (**f**) mice had more connective tissue (cardiac muscle fibres) than WTCD (**d**) and KOHFD (**e**). Magnification = 20×. Scale bar = 200 µm.

**Figure 8 biology-11-00311-f008:**
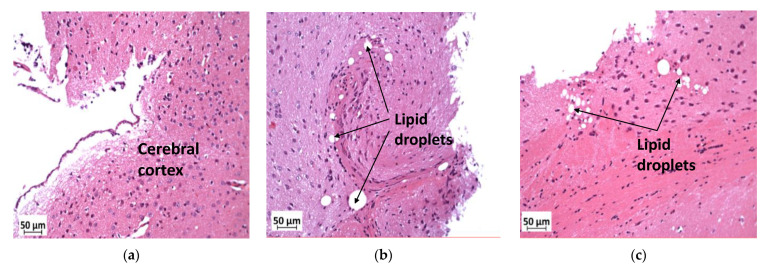
Effect of *slc7a8* deletion on lipid droplet accumulation in brain tissue. H&E-stained sections of brain tissue showed lipid droplets in the cerebral cortices of KOHFD (**b**) and WTHFD (**c**) that were not seen in that of WTCD (**a**). Magnification = 20×. Scale bar = 50 µm.

**Figure 9 biology-11-00311-f009:**
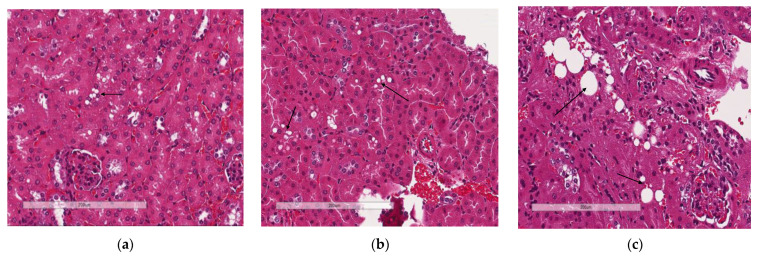
Effect of *slc7a8* deletion on lipid accumulation in the kidneys. H&E-stained sections showed that accumulation of lipids (black arrows) was greater in WTHFD (**c**) than in WTCD (**a**) and KOHFD (**b**). Magnification = 20×. Scale bar = 200 µm.

**Figure 10 biology-11-00311-f010:**
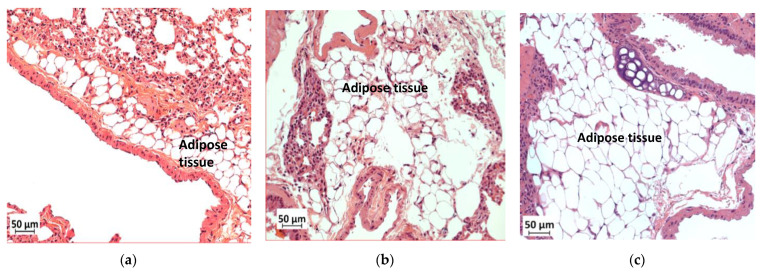
Effect of slc7a8 deletion on lipid accumulation in the lungs. H&E-stained lung sections showed the accumulation of adipose tissue in the lungs, which was greater in WTHFD (**c**) than in KOHFD (**b**) and WTCD (**a**). Magnification = 20×. Scale bar = 50 µm.

**Figure 11 biology-11-00311-f011:**
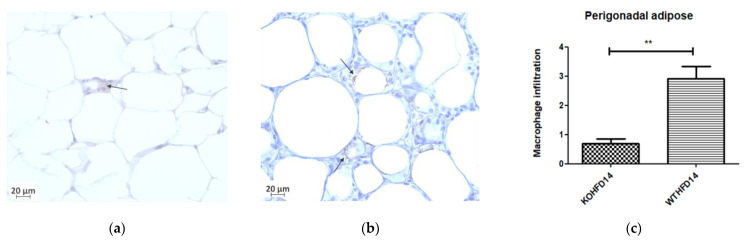
Effect of *slc7a8* deletion on macrophage infiltration in adipose tissue. KOHFD (**a**) showed a significant decrease in macrophage infiltration (indicated by black arrows) in pWAT (*p* < 0.01), (**c**) compared to WTHFD (**b**). (**d**,**e**) represent the negative controls for KOHFD and WTHFD, respectively, in perigonadal adipose tissue. A significant decrease in macrophage infiltration in mWAT (*p* < 0.05) (**h**) was observed in KOHFD (**f**) compared to WTHFD (**g**). A significant decrease in macrophage infiltration in iWAT (*p* < 0.01) (**k**) was observed in KOHFD (**i**) compared to WTHFD (**j**). No significant differences in macrophage infiltration in brown adipose tissue (**n**) were observed between KOHFD (**l**) and WTHFD (**m**). Magnification = 40×. Scale bar = 20 µm. N = 5 fields. * *p* < 0.05, ** *p* < 0.01.

**Figure 12 biology-11-00311-f012:**
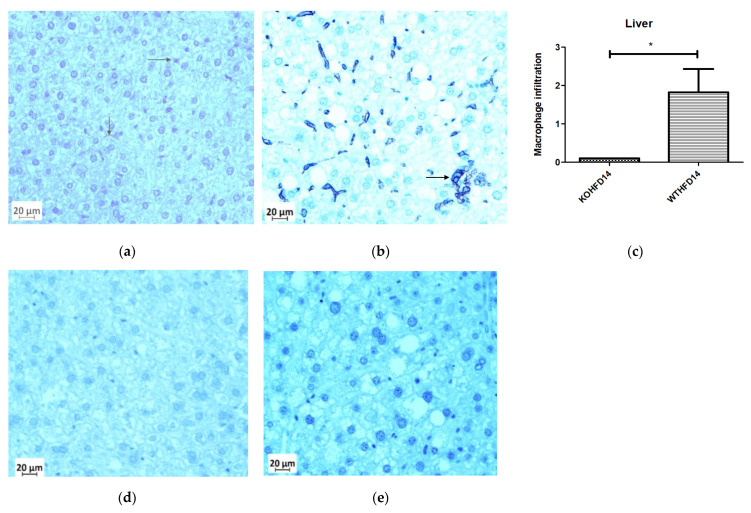
Effect of *slc7a8* deletion on the presence of macrophages in the liver. WTHFD (**b**) had significantly greater infiltration (*p* < 0.05) (**c**) of macrophages than KOHFD, (**a**). Negative controls for KOHFD (**d**) and WTHFD (**e**); no macrophages were detected. Magnification = 40×. Scale bar = 20 µm. N = 10 fields. * *p* < 0.05.

**Figure 13 biology-11-00311-f013:**
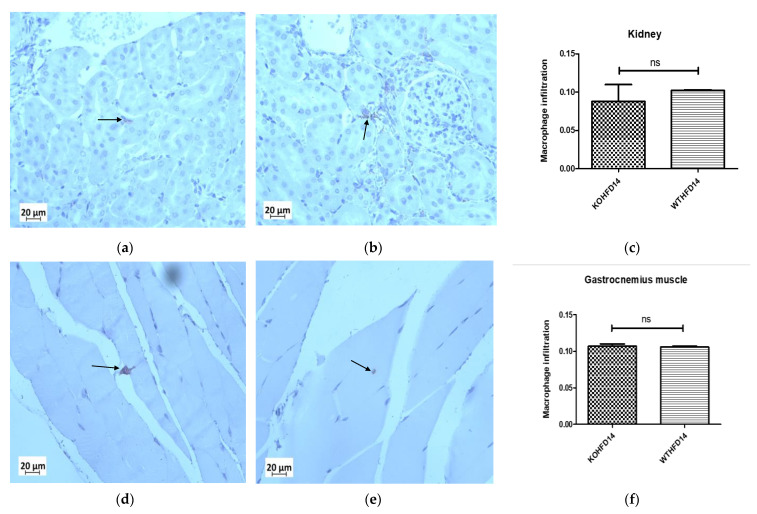
Effect of *slc7a8* on macrophage infiltration profiles of the kidney and gastrocnemius muscle. KOHFD (**a**) had slightly fewer macrophages infiltrating into the kidney (**c**) than WTHFD (**b**). However, no significant differences were noted between KOHFD and WTHFD. No significant differences (**f**) were observed in infiltration between the gastrocnemius muscles of KOHFD (**d**) and WTHFD (**e**). Magnification = 40×. Scale bar = 20 µm. N = 5 fields.

## Data Availability

Not applicable.

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
