# Peer review of "Slc7a8 Deletion Is Protective against Diet-Induced Obesity and Attenuates Lipid Accumulation in Multiple Organs"

_biology, 2022, doi:10.3390/biology11020311_

Round 1
Reviewer 1 Report
In this manuscript, the authors focused on the function of SLC7A8 (Solute Carrier Family 7, Member 8) on diet induced obesity, and demonstrated that the SLC7A8 gene deficient mice are resistant to obesity caused by high fat diet. The authors also shown that the lipid accumulation various organs and tissues caused by high fat diet was reduced in the SLC7A8 gene deficient mice compared to the wild type mice. On the other hand, most part of this study is histochemical analysis by haematoxylin and eosin (H&E) staining, and this study lacks serological analysis (e.g. serum total cholesterol and serum triglycerides levels) and lipid staining (e.g. Oil Red O staining). Therefore, more detailed analysis would be required to understand the SLC7A8 gene function on obesity. Further, the data shown in this study is difficult to understand whether SLC7A8 gene is crucial for the lipid accumulation in adipocytes, or other factor involved in obesity. However, the findings of this study seem to be novel, and I believe that the results shown this paper are interesting for the readers of the journal, Biology.
Specific point:
The authors should provide the data for total food intake in each group of mice.
Reviewer 2 Report
Reabetswe R. Pitere and colleagues in their manuscript entitled ‘Slc7a8 deletion is protective against diet-induced obesity and 2 attenuates lipid accumulation in multiple organs’ extensively evaluated the importance of slc7a8 in different tissues using a mouse model. This is an interesting study with several tissues revealing the deficiency of the Slc7a8. However, I have important concerns and comments to improve the quality of the manuscript.
- In the introduction, the entire background information to set up the hypothesis reads very lengthily.
- Authors should provide more background on why they choose SLC7A8 and reveal its known roles, instead, authors stressed too much on obesity background.
- At the end of the introduction, please remove the sentence ' in some of these tissues' and replace it with what are those exact tissues.
- In the entire manuscript, authors should change the abbreviations to actual terms, not WTHFD and KOHFD.
- For the methods section, please provide detailed kit and catalogs number along with the company name, also provide Jax mice details with numbers
- Was the organ weight normalized to bodyweight?
- KO control diet is missing in the adipose-specific data. Scale bars are not clear on stained images, please provide them in a different color to appreciate the size in each image.
- In fig1a and fig1c, the authors should explain why there is decreased body weight in WTCD, while there is an increase in the energy intake (for example, Fig1c)?
- Clearly label the fig2a data if GTT is from the KO mice or Control mice and the diet details in the image. The figure legend is misleading with CD and HFD.
- Fig 4, adipocyte diameter images look cropped, please provide two different representative images in the full microscopic field area.
- To strengthen the confidence in the results of the liver lipid accumulation data, authors should add liver function test details and provide AST-ALT levels.
- Fig 6 again, KO control diet data is missing. For Fig 11, please provide negative staining control data.
Reviewer 3 Report
The manuscript provides very interesting results that are obtained from in vivo experiment. Moreover, the results and their discussion are presented in good style. However, I have a minor point that is required to be addressed and clarified.
It would be better to discuss the undesirable effects of Slc7a8 deletion. This point is very important since Slc7a8 deletion is not practically applied in clinical practice.
Reviewer 4 Report
Dear Editor, thank you for inviting me to revise this manuscript.
Unfortunately, this manuscript is too unstructured to be submitted, so Major revision will not be able to revise it, and it will most likely be rejected after author's REVISE round.
It is in the best interest of the authors, biology, editors, and reviewers to resubmit the entire manuscript, including figures, from scratch rather than doing a Revise.
Therefore, I consider to be "reject".
- Please explain more about the SLC7A8 gene in the introduction section.
It is insufficient to provide information to the reader. - I am concerned that the scale of Figures 4, 5, 6, 7, 9 is uneven.
And, also, the scale bar is difficult to see or missing in all the diagrams, so they are not appropriate as diagrams.
The arrows, circles, and text in the figures are difficult to read and not up to the level required by the biology.
Please unify the photos to scale, and/or provide a diagram that more clearly shows the scale bar in the image.
As it is, it is impossible to determine if the results are the result of an accurate analysis. - There are too many defects in the description of the "Reference".
The most notable are numbers 24 and 25.
and more...
Round 2
Reviewer 2 Report
This manuscript is improved after revision and may be acceptable for publication.
Author Response
Comment noted. Thank you.
Reviewer 4 Report
This revised manuscript has not yet been formatted as an academic paper.
・The resolution of the images on the lower side of Fig4 A to C, both of Fig. 4 K and L, the upper side of Fig. 4 P ~ R, and the upper side of Fig. 5 D ~F is extremely poor and I do not know what they represent.
・Furthermore, Fig4Q sees the same field of view of more than 70%. It is unfair to submit this as an image of another field of view.
・I don't even know what is written inside the scale bar in Fig6. The point that the scale bar cannot be seen has not been resolved. If it cannot be solved, this photo does not conform to the quality of "Biology".
・In addition, the fact that the scales are not unified in the photographs for which diameters and numbers should be compared in Fig. 4 K to L may be misleading to the reader. Replace it with an image of the same scale to show that it is not an arbitrary scale change.
・This should be applied to each of the figures in this manuscript.
・In Fig. 8C etc., the scale bar is interrupted.
Originally, peer review is conducted based on a relationship of trust.
However, the authors only provide unreliable figures. I can't trust this author on the following points:
Therefore, I will "reject" this manuscript again.
1) Remarkably low resolution
2) Images to be compared and contrasted are not set on the same scale
3) Fatal flaws in the scale bar (visibility and proper notation)
4) Submit almost the same field of view as another field of view
